# Effect of Picloram and Desiccation on the Somatic Embryogenesis of *Lycium barbarum* L.

**DOI:** 10.3390/plants13020151

**Published:** 2024-01-05

**Authors:** Poonam Khatri, Nirmal Joshee

**Affiliations:** Agricultural Research Station, Fort Valley State University, Fort Valley, GA 31030, USA; fnu.poonam@fvsu.edu

**Keywords:** desiccation, goji berry, plant growth regulators, somatic embryogenesis

## Abstract

An efficient and reproducible in vitro method for indirect somatic embryogenesis was optimized by culturing leaf and leaf with petiole explants of *Lycium barbarum* L. Murashige and Skoog (MS) medium, supplemented with various concentrations of Picloram and 2,4-Dichlorophenoxyacetic acid (2,4-D), individually and in combinations, were tested. Picloram (1.0 µM) showed a better response compared to 2,4-D and results indicate it to be a better auxin for induction of somatic embryos for Goji berry. It was seen that the leaf explants were more responsive in callus and somatic embryo induction than the leaf with petiole explant when incubated in the dark for 5 weeks. Embryogenic callus, after being transferred to MS medium containing Benzyl amino purine (BAP) in 1.0 µM, 5.0 µM and 10.0 µM, began to differentiate in light after one week. MS medium with 1.0 µM Picloram + 10 µM BAP resulted as the most favorable treatment for somatic embryogenesis in *Lycium barbarum* L. Removal of plant growth regulators from MS medium and culturing induced calluses under 16 h photoperiod resulted in globular, heart, torpedo, cotyledons, and further development into plantlets. Well-developed plants have been obtained and are capable of acclimatizing in ex vitro conditions. In addition, the effects of desiccation treatments (0, 1, 3, 6, 9 h, and 12 h) on embryogenic callus for somatic embryo induction were found to be directly proportional to the length of desiccation treatment at room temperature. After 9 h and 12 h of desiccation treatments, 60% and 90% of plated calluses resulted in somatic embryos, respectively. In a *L. barbarum* callus mass, Acetocarmine and Evans blue double staining differentiated between embryogenic and non-embryogenic callus. These findings will help Goji berry improvement by elite clone production, ex situ conservation projects, scaling up plant production, and agronomy for the commercial production of this superfruit in the future.

## 1. Introduction

Plant-based products are gaining attention in the medicinal, nutraceutical, and cosmeceutical industries. One such plant used for centuries in Eastern traditional medicine is *Lycium barbarum* L., which bears superfruit status and is known as ‘red diamonds’ [1].

Goji berries (*Lycium* fruits) are harvested from two closely related plants, *Lycium chinense* and *Lycium barbarum*. The *Lycium* genus belongs to the Solanaceae family and has been used in Chinese traditional medicine for centuries using berries, leaves, and root bark. There are about 100 species of *Lycium* distributed in North America (24 species), South America (32 species), South Africa (24 species), Eurasia and Africa (2 species) and temperate Europe and Asia (12 species) [2,3]. Goji (*Lycium barbarum* L.) is a fruit-bearing woody shrub that has become increasingly popular in recent years due to its nutritive and antioxidant properties [4] and several other health benefits, such as decrease in blood lipid concentration, improved immunity, promoting fertility, and benefits for kidney, liver, and vision [4,5,6].

Polysaccharides are the most important compound present in goji berries, which are attributed to have antioxidant, anti-inflammatory, cardio-protective, hepatoprotective, hypoglycemic and immune system boosting activities. *L. barbarum* polysaccharide (LBP) reduces tumor growth by causing apoptosis and cell cycle arrest. It has been found that LBP can also suppress the carcinoma cell growth in human bladder cancer [7]. Carotenoids, lignans, phenylpropanoids, phenolic acids, flavonoids, coumarins, and alkaloids are secondary metabolites that have been reported from goji berries [4,8]. Goji berries are commercially placed under the ‘Superfruit’ category and are consumed as functional foods, nutraceuticals, and beverages, etc. Superfruits have a high nutritional value due to their richness in antioxidants, nutrients, potential health benefits, and good taste [1].

Goji berry has been propagated traditionally through seeds that suffer from low germination, lack of cell proliferation, inconsistent crop yield and unpredictable crop growth [9]. Therefore, it is important to develop and optimize a rapid production system by in vitro culture to produce desirable plants with high medicinal properties. Many of the research papers are on in vitro plant regeneration through shoot morphogenesis using 2,4-D (2,4-Dichlorophenoxyacetic acid), BAP (6-Benzyl aminopurine), and NAA (α-Naphthaleneacetic acid). However, there are few research studies on callus-mediated somatic embryogenesis and plant regeneration (Table 1). Somatic embryogenesis (SE) is considered the most suitable in vitro method for the clonal propagation of different plant species due to its high multiplication rate. It can be used for the propagation of an elite genotype, and it is one of the most important prerequisites for genetic interventions [10]. Several researchers have reported on plantlet regeneration by organogenesis as well as somatic embryogenesis. Somatic embryogenesis is a developmental process by which somatic cells of plants produce embryogenic cells through a sequence of morphological and biochemical changes to form a somatic embryo under suitable inductive conditions [11]. Somatic embryogenesis presents a suitable system for their controlled production and observation and collection of various developmental stages under in vitro conditions for biochemical and molecular analyses [10].

A well-established protocol for plant regeneration of *L. barbarum* may offer new possibilities for genetic transformation, large-scale multiplication, and conservation of this elite germplasm. This research was conducted to develop a reproducible and rapid protocol for the micropropagation of *L. barbarum* via somatic embryogenesis from the leaf and leaf with petiole explants.

## 2. Results and Discussion

### 2.1. Effects of Explants and Plant Growth Regulators on Callus Induction and Somatic Embryogenesis

In initial experiments, MS medium with high concentrations of auxins (20 µM to 40 µM) caused browning and produced a non-embryogenic callus. A second set of experiments were carried out with 0.1 µM to 10.0 µM of auxins (Table 2). It was observed that leaf explants were more responsive in callus and somatic embryo induction than leaf with petiole explants when incubated in the dark for 5 weeks (Table 2). The highest percentage of embryogenic callus in leaf explant was 92% and 18% was the lowest shown by the leaf with petiole (Figure 1).

In the current study, callus formation initiated at the cut end of the explant after 7 days (dark) in almost all the treatments containing MS Medium with Picloram and 2,4-D, individually and in combinations (Figure 2A,B). Callus formation was observed in all treatments, but there were variations in the frequency and characteristics of the callus formed on different PGRs, individually or in combinations. No callus formation took place in the control. Friable, nodular, and yellowish callus (embryogenic callus) from culture treated with auxin was selected for a study of somatic embryogenesis and plantlet regeneration in *L. barbarum*. Only one nutrient formulation (MS medium) was used in all the experiments. Somatic embryogenesis may progress on a single medium or may require several medium alterations specific for a developmental stage [26].

The choice of explant is a critical factor in the induction of somatic embryogenesis, as suggested in the literature. Genotype, age of explant and source of explant can affect the induction of somatic embryogenesis [27]. It is reported that the different explants from the same mother plant may respond differently in callus, embryogenic callus production, or any other morphogenic response. The amount of endogenous phytohormones in the explants, exogenously supplied type, concentrations of cytokinin and auxins used, and their uptake and interaction with endogenous plant growth regulators are some of the factors that decide a particular outcome [27]. Studies have shown that exogenous supply of auxin can increase the endogenous auxin level, which may lead to the establishment of a hormonal gradient ideal for the induction of embryogenesis [27,28]. In the case of *Centella asiatica* L., it was found that the physiological status of explant is important for somatic embryogenesis [28]. A similar response in *L. barbarum* was observed, where type of explants and plant growth regulators exert synergistic effects on callus formation [22].

The role of Picloram on somatic embryogenesis has been studied for the first time and identified as an effective auxin in *L. barbarum* (Table 3). Typically, 2,4-D has been the most used auxin for somatic embryogenesis [22,24]. Induction of somatic embryos on a medium containing Picloram has been reported in strawberry (*Fragaria ananassa* L.) [29]. In a study comparing Picloram and 2,4-D for induction of somatic embryogenesis in wheat, barley and Tritordeum, embryogenesis was achieved in medium containing Picloram. In addition, it was stated that callus induction was two times more efficient under Picloram as compared to 2,4-D [30]. Osman et al. [22] and Verma et al. [24] achieved somatic embryogenesis in Goji berry with the supplementation of 2,4-D alone.

Embryogenic callus, after being transferred to MS medium containing BAP (1.0 µM, 5.0 µM and 10.0 µM), began to differentiate in light after one week. MS medium with 1.0 µM Picloram + 10.0 µM BAP resulted as the most favorable treatment for somatic embryogenesis at various developmental stages after 6 weeks in light (Figure 2C) and the mean of the number of somatic embryos is provided in Table 3.

Under a dissecting microscope, embryogenic callus incubated on basal MS medium for six weeks under 16 h photoperiod revealed various developmental stages (Figure 2C–H). The use of auxins with cytokinin has been reported prior for somatic embryogenesis [22,31]. In general, auxin and cytokinin and their relative concentration is the key factor responsible for the embryogenic response. This could be due to their involvement in the regulation of cell cycle, cell division, and differentiation [31]. It was found that 0.3 mg/L 2,4-D and 0.1 mg/L BAP were the best treatment to produce somatic embryos using the leaf and nodal segment of *L. barbarum* [22]. In the present study, repetitive somatic embryogenesis was frequent and led to the formation of many somatic embryos. Secondary embryogenesis was also observed on the developing embryos (Figure 2I). In literature, secondary somatic embryos are also known as recurrent, cyclic, adventitious, or repetitive somatic embryos [32]. This may increase and extend the embryogenic competence of certain lines, can multiply the number of embryos, and reuse somatic embryos of abnormal morphology that otherwise cannot regenerate into normal plants [32].

### 2.2. Somatic Embryo Development, Germination, and Plantlet Regeneration

To promote further development and germination of somatic embryos in goji berry, PGR free MS medium was tested. At this stage, somatic embryos were germinating and progressing through various stages (globular, heart, torpedo, and cotyledon) of somatic embryogenesis on PGR free MS medium. In the present study, somatic embryos followed an asynchronous mode of development due to the presence of different development stages in the same medium (Figure 3A,B). This has been reported in other plants, such as *Centella asiatica* L. [28] and *L. barbarum* [19]. Subculturing of embryogenic callus to the PGR-free medium in presence of light has increased the frequency of somatic embryos. Adventitious shoots appeared from the green mass (Figure 3A) and subsequent subcultures on same media enhanced the rate of shoot proliferation.

The transfer of somatic embryos to PGR free medium changed the morphological appearance and leaf primordia were formed. Somatic embryos further differentiated into plantlets on plant growth free MS medium under a 14 h photoperiod (Figure 3C). Auxins play an important role in the callus induction and rooting but usually inhibit plantlet regeneration [33]. It is suggested that somatic embryos should be subcultured to a PGR-free medium for the plantlet formation [33]. The use of PGR-free medium has been found useful in somatic embryo maturing and plantlet regeneration in many other species, such as triticale [33], eucalyptus [34] and *Vicia faba* [35]. A fully developed normal embryo is shown in (Figure 2G). Further extension of epicotyl and shoot is evident, along with the root growth (Figure 2H).

### 2.3. Abnormal Embryos

Normal and abnormal embryos were found to occur in the same medium. Long-term culture at high concentration of Picloram and BAP has resulted in abnormal embryos such as multiple cotyledons, lack of apical and radical meristems, fusion of two or more embryos and leaf like structures. Abnormal embryos show poor regeneration into a complete plantlet. Similar results have been reported in *L. barbarum* and *Centella asiatica* L. [19,28]. Researchers have noticed that a high concentration of auxins can disturb the normal genetic and physiological activities in cells. Further, prolonged exposure and accumulation of exogenous auxins inside the tissues can affect the normal somatic embryo development [36].

### 2.4. Acclimatization

When plantlets with well-developed roots reached about 3 cm long, plantlets with 4–5 leaves were randomly selected, washed gently to remove agar, and then transferred randomly to the climate-controlled acclimatized boxes (Smither-Oasis, Kent, OH, USA) having sterile Promix (BX, Pittsburgh, PA, USA) (Figure 4). A total of 80 plantlets (20 plantlets from each treatment) were transferred to the acclimatization boxes. There were no morphological abnormalities observed in the regenerated plantlets, showing 100% survival in green house conditions (at 65–70% relative humidity and 27 ± 2 °C). Other authors have reported variable responses in terms of acclimatization success, ranging from 80 to 95% in *L. barbarum* [19,21].

### 2.5. Cytochemical Staining

For cytochemical analysis of callus, double staining was performed to differentiate between embryogenic and non-embryogenic callus. For this study, we have selected two types of calluses—green embryogenic callus with high regenerative potential at later stages in light (Figure 5A) and brown non-embryogenic callus in light (Figure 5D). In vitro, callus cultures are composed of two types of cells. The embryogenic cells are rounded, densely cytoplasmic and are reactive to Acetocarmine (red) while non-embryogenic cells are vacuolated, elongated, and permeable to Evans blue (blue) in Figure 5. Cell morphology and embryogenic potential have not been studied before in *L. barbarum*. It has been reported that the cell morphology and embryogenic potential can be discovered by double staining [37,38]. Double staining protocol using Acetocarmine (0.1%) and Evans blue (2%) was used to distinguish between embryogenic and non-embryogenic cells. Acetocarmine is used to detect DNA, chromatin and macromolecules present in viable embryogenic cells. Evans blue can penetrate through the ruptured membrane and can stain non-viable non-embryogenic cells [37,38].

### 2.6. Effect of Desiccation Treatment on Somatic Embryo Induction

To evaluate the potentiality of the embryogenic callus response to desiccation, 4 weeks old callus were exposed to various desiccation treatments (0, 1, 3, 6, 9, and 12 h) and then cultured on MS medium with 1.0 µM Picloram added. The results revealed that the percentage of somatic embryos produced was significantly different between the desiccation treatments (Figure 6).

After 9 h and 12 h of desiccation treatments, 60% plated calluses and 90% plated calluses resulted in somatic embryos, respectively (Figure 7). However, there was no induction of somatic embryos reported after 0 h desiccation. The results showed that calluses plated on 1.0 µM Picloram, after 1 h desiccation, yielded somatic embryos on 35% of calluses, which increased to 37% for 3 h and 55% for 6 h desiccation treatment after 5-week incubation, in comparison to the controls.

In the present study, the desiccation treatment played an effective role in inducing the somatic embryos. This is the first report of desiccation on somatic embryo induction in *L. barbarum*. Our results agree with Siddique et al. [39] on the desiccation effect in three rice genotypes strongly influencing the regeneration frequency up to 2- to 4-folds. It has also been reported that regeneration frequency is 2–4 folds higher from 3 h desiccated callus than non-desiccated callus [40]. The studies have reported that desiccation treatment could be dependent on water content in the cells of the callus. Callus of lower age might have a large amount of water and hence need longer duration of desiccation for maximum regeneration while mature callus may require desiccation for shorter duration to produce somatic embryo induction. It has also been noticed that regeneration may be dependent on other factors like genotype, age of the callus and duration of desiccation [39]. It has also been reported that callus exposed to longer desiccation treatment suppressed the growth of the embryos, as well as the plantlet regeneration [41]. Therefore, we plan to further investigate the effect of desiccation treatment on somatic embryogenesis and plant regeneration, considering the age of callus, duration of desiccation beyond 12 h and various explants in our future investigation.

## 3. Materials and Methods

The following section describes plant material used and protocols followed for the induction and proliferation of embryonic callus, and maturation and regeneration of somatic embryos.

### 3.1. Plant Materials

Actively growing leaf and leaf with petioles of Goji berries were collected from the greenhouse-grown plants at Fort Valley State University, Georgia, USA. Plant material was washed under running water for 10 min and then washed for another 45 min in distilled water with 2% FungigoneTM (Plant media) and 2–3 drops of tween-20 (Sigma, St. Louis, MO, USA) with constant stirring. These plants were further rinsed in sterile distilled water to remove traces of Fungigone and tween-20 in the laminar air flow cabinet, and then dipped in 70% ethanol for 1 min followed by one rinse in deionized water. Final sterilization was carried out by immersing plant material in 15% commercial bleach Clorox (Oakland, CA, USA) for 10 min with constant stirring followed by three washes of 5 min each in sterile distilled water [26].

### 3.2. Media Preparation and Culture Conditions

Murashige and Skoog (MS) basal medium were supplemented with 3.0% sucrose as the carbon source and TC Agar (0.7%) as the solidifying agent. The pH of the medium was adjusted to 5.8 before adding agar and plant growth regulators. A medium devoid of any plant growth regulators was used as control. The culture media was dispensed in petri dishes (90 × 15 mm) and each dish had 25 mL medium. To find the most suitable explant for somatic embryo induction, 20 explants were used for each treatment and each experiment was repeated three times. All the cultures initiated for somatic embryogenesis were incubated in the dark at 25 °C ± 2 °C.

### 3.3. Induction of Embryogenic Callus and Somatic Embryo Formation

The MS medium was supplemented with several concentrations (0.1 µM to 10.0 µM) of Picloram and 2,4-D individually and in combinations to investigate the role of auxins in somatic embryo induction. Various stages during somatic embryogenesis of *L. barbarum* are presented in Table 4.

### 3.4. Cytochemical Analysis of Callus

A small piece of callus (50 mg) was isolated, placed on a slide, and stained with 2% Acetocarmine until the callus was submerged completely. Callus was gently divided with forceps into small pieces and then rinsed with distilled water 2–3 times. Afterward, the samples were stained with 0.5% (*w*/*v*) Evans Blue with Acetocarmine-stained cells for 30 s and washed again with distilled water to wipe off extra dye. Then, 1–2 drops of glycerol were added to the stained cells and mounted with the coverslip. Callus cells were then examined under the microscope (Olympus BX 43, Olympus, Bartlett, TN, USA). Double staining with Acetocarmine and Evans blue was used to differentiate between embryogenic calluses and non-embryogenic calluses. The embryogenic cells are reactive to Acetocarmine and stains red while the non-embryogenic cells are permeable to Evans Blue dye and show a blue color [35,37].

### 3.5. Effect of Desiccation Treatment on Somatic Embryo Induction

Effect of desiccation on induction of somatic embryogenesis was studied at an interval of 0, 1, 3, 6, 9, and 12 h. Four-week-old callus from the dark was subjected to different desiccation treatments. The desiccation treatment was employed by placing the callus in petri plate opened under Laminar Air Flow for a period of 0, 1, 3, 6, 9 and 12 h prior to culture on MS medium with 1.0 µM Picloram. Approximately, 10 calluses (100 mg each) per petri plate were used. These desiccated calluses were incubated at 25 ± 2 °C under a 16 h photoperiod at 60 μmol m^−2^ s^−1^. At the end of the experiment, the percentage of green spot formation obtained was recorded after 5 weeks.

### 3.6. Acclimatization, Hardening, and Greenhouse Transfer of Plants

The regenerated plants through somatic embryogenesis were gently removed from the culture vessel and roots were carefully washed to eliminate all the gel and media remains. Later, these plants were transferred into humidity control boxes (Smithers-Oasis, Kent, OH, USA) having Promix BX, Premier Horticulture Inc., (Quakertown, PA, USA) and incubated at 25 ± 2 °C 16 h light 60 μmol m^−2^ s^−1^ for 21 days (about 3 weeks). The acclimatized boxes used in our study have a humidity control filter assisted wheel regulating humidity exchange that remained closed for 14 days and then the humidity was slowly released by opening the wheel gradually every day for 7 days. Thereafter, hardened plants with new growth were transferred to larger pots with a perlite and potting mix (1:3 ratio), moistened with tap water and maintained in the green house for further growth. Plants were grown to maturity under greenhouse conditions at 65–70% relative humidity and 27 ± 2 °C.

### 3.7. Statistical Analysis

Somatic embryogenesis Induction experiments were set up with 20 explants for each treatment and each experiment was repeated three times. Data were expressed as mean ± SE of the triplicates. Data were statistically processed by analysis of variance (ANOVA) and means were compared by Duncan’s Multiple Range Test (DMRT) at 0.05 level of probability using SPSS analytic software (IBM SPSS Statistics version 22.0 (stats.exe).

## 4. Conclusions

The current study reports a new, reproducible, and rapid method for in vitro multiplication of *Lycium barbarum* L. To date, there have been no reports on the role of Picloram and desiccation on the induction of somatic embryos in Goji berry. Our studies indicate that the leaves harvested from greenhouse grown plants were suitable material for producing embryogenic callus and half leaf explant was more responsive than the leaf with petiole when incubated in the dark for 5 weeks. In the present study, Picloram was found to induce a better response compared to 2,4-D for the induction of somatic embryos in *L. barbarum*. Further, Acetocarmine and Evans blue staining could help in differentiating embryogenic callus from non-embryogenic callus. Four-week-old callus grown in the dark were plated on MS fortified with 1.0 µM Picloram after 12 h desiccation, resulting in 90% of calli producing somatic embryos after 5-week incubation while no somatic embryos were seen on control calli not subject to desiccation. Further investigations will be needed to optimize the most suitable desiccation treatment for mass multiplication. This protocol may further aid in the mass propagation of this superfruit-bearing shrub to meet demand and can serve as a basis for a long-term germplasm conservation strategy.

## Figures and Tables

**Figure 1 plants-13-00151-f001:**
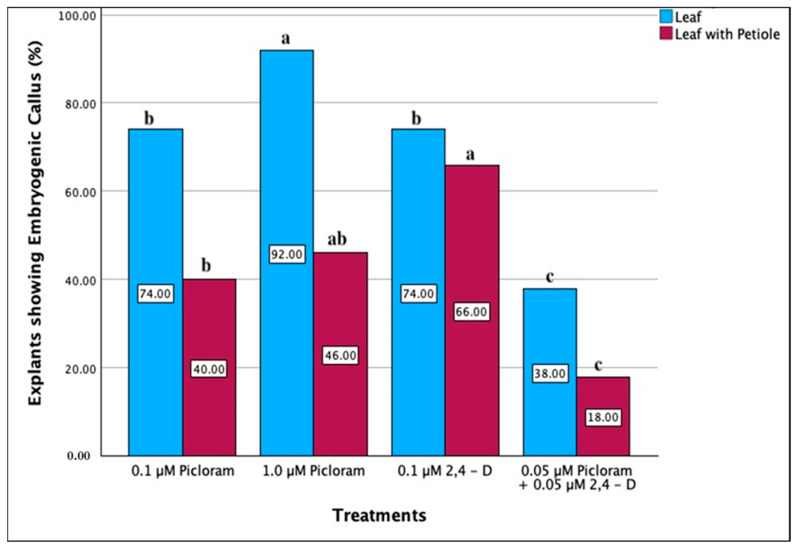
Effect of picloram and 2,4-D concentrations on the induction of embryogenic callus in leaf and leaf with petiole explants. Data for treatments that produced embryogenic response from either leaf or leaf with petiole has been shown. Each value represents mean, and values with the same superscript are not significantly different at 5% probability level according to DMRT.

**Figure 2 plants-13-00151-f002:**
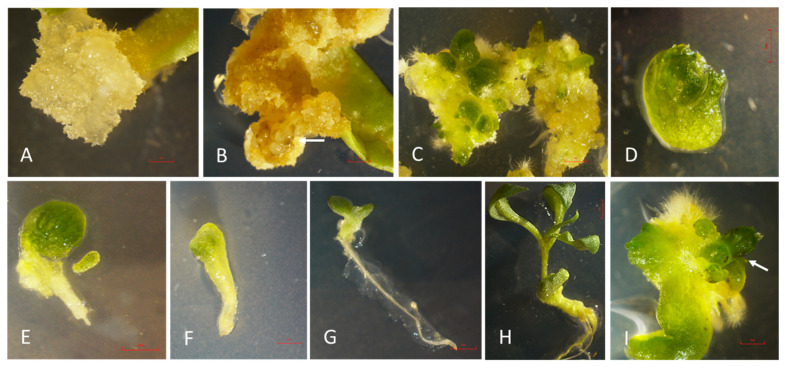
Somatic embryogenesis in *L. barbarum*. (**A**) Callus induction at the cut end of leaf explant in response to MS medium with 1.0 µM Picloram (0.63×). (**B**) Induction of nodular callus in leaf explant on MS medium with 1.0 µM Picloram (0.63×). (**C**) Callus greening, development of embryos, and formation of root/shoot axis on MS medium with 10.0 µM BAP (Light) (0.63×). (**D**–**F**) Developmental stages during somatic embryogenesis on the MS medium with 10.0 µM BAP (Light) (1×). (**G**) An isolated germinating embryo with shoot and root formation (0.63×). (**H**) First true leaf development, and internode elongation (0.63×). (**I**) Secondary somatic embryos on the main embryo in the PGR-free MS medium (0.63×). Arrows indicate nodular callus and secondary somatic embryos in (**B**) and (**I**), respectively. Pictures were taken using a microscope (Nikon Stereo Microscope SMZ1270, Melville, NY, USA). Scale bars: 2 mm (**A**–**I**).

**Figure 3 plants-13-00151-f003:**
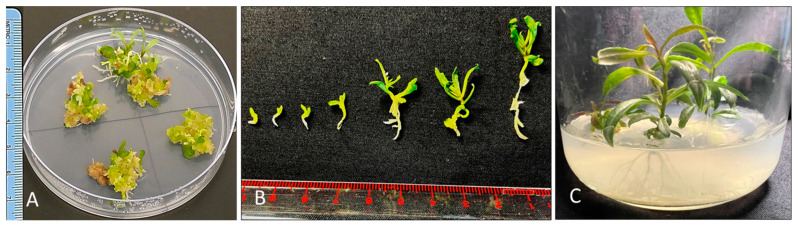
Progressive development of somatic embryos. (**A**) Clusters of somatic embryos in different developmental stages (**B**) Germinating somatic embryos and elongation. (**C**) Well-developed shoots and roots formed on PGR free MS medium. Diameter of the culture jar is 60 mm.

**Figure 4 plants-13-00151-f004:**
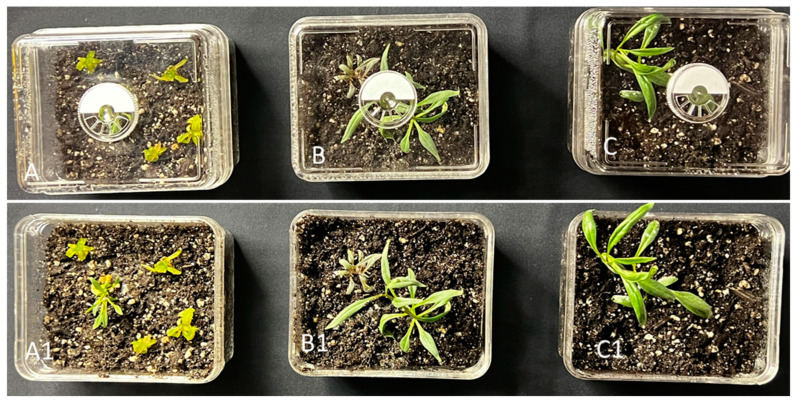
Rooting and plant acclimatization in humidity regulated boxes. Well-rooted plantlets developed from somatic embryos were hardened in boxes containing sterilized potting mix (Promix BX). (**A**–**C**). Acclimatized plantlets in controlled humid conditions. (**A1**–**C1**) Acclimatized plants in the greenhouse conditions.

**Figure 5 plants-13-00151-f005:**
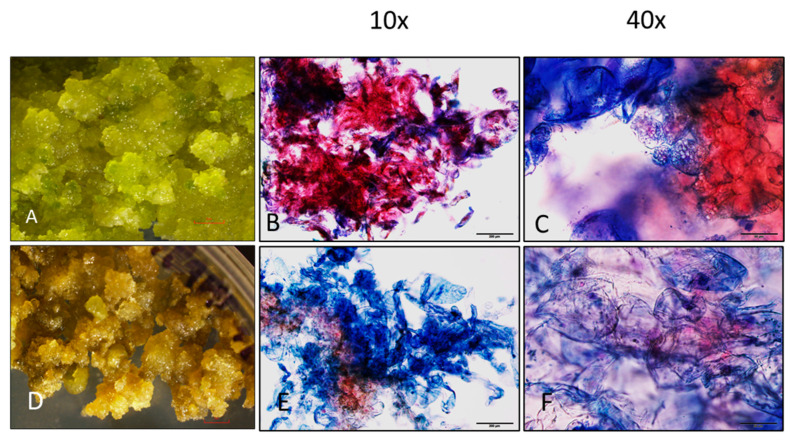
Evans blue (EB)-Acetocarmine (AC) staining of callus to differentiate embryogenic (red) and non-embryogenic callus (blue) in *L. barbarum*. (**A**) Green embryogenic callus with high regenerative potential (0.63×). (**B**,**C**) Non-embryogenic and embryogenic cells are differentiated as blue and red (10× and 40×, respectively). (**D**) Brown non-embryogenic callus (0.63×). (**E**) Loose structure of the cell mass (10×). (**F**) Elongated and vacuolated cells as in (**E**) at a higher magnification (40×). (**A**,**D**) Pictures were taken using Nikon Stereo Microscope SMZ1270, Melville, NY, USA. (**B**,**C**,**E**,**F**) Pictures were taken using Olympus BX 43, Olympus, Bartlett, TN, USA. Scale bars: 2 mm (**A**,**D**); 200 μm (**B**,**E**); 50 μm (**C**,**F**).

**Figure 6 plants-13-00151-f006:**
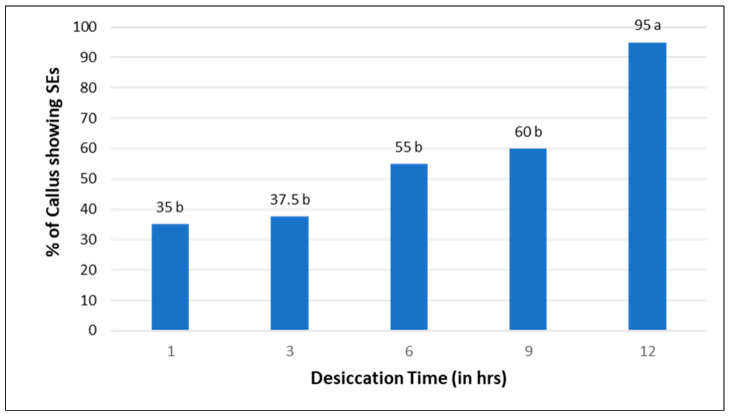
Effect of desiccation on the induction of somatic embryos. Each value represents the mean, and values with the same superscript are not significantly different at a 5% probability level according to DMRT.

**Figure 7 plants-13-00151-f007:**
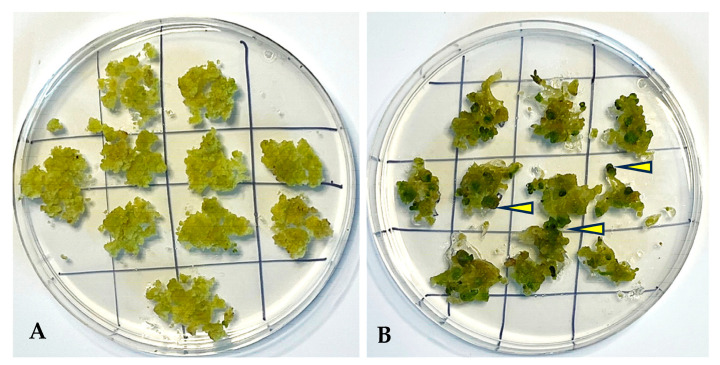
Role of desiccation in somatic embryo induction from callus. (**A**) Callus prior to desiccation treatment. (**B**) Somatic embryos appeared after 5 weeks in 12 h desiccation treatment. The arrows point to the dark green somatic embryos.

**Table 1 plants-13-00151-t001:** Review of in vitro propagation protocols for plant regeneration in *L. barbarum.*

Explants	Media Compositions	Plant Regeneration	References
Leaf and young internodes	MS with 0.2 mg/L NAA and 1 mg/L 2,4-D MS: Murashige and Skoog [12]	Shoot Organogenesis	[13]
Leaf, callus, and callus protoplast	B5 medium [14] containing 1.5 mg/L (BAP) and 0.5 mg/L NAA	Shoot Organogenesis	[15]
Leaf	MS medium supplemented with 2 mg/L BA and 0.5 mg/L NAA	Shoot Organogenesis	[16]
Leaf	MS medium 1 mg/L IBA (Indole 3 Butyric Acid) and 1 mg/L IBA IAA	Somatic Embryogenesis	[17,18]
Root	MS medium containing 0.2 mg/L 2,4-D	Indirect Somatic Embryogenesis	[19]
Nodal Segment	DKW (Driver and Kuniyuki Walnut) medium supplemented with 6-benzylaminopurine (BAP; 0.5 mg/L) and sucrose 3% *w*/*v*	Micropropagation and Ex vitro Rooting	[11]
Leaves, Apexes and Nodal segments	Liquid MS Medium with 1 mg/L IBA	Shoot Organogenesis	[20]
Leaf and nodal segments	MS Medium with 1.33 and 2.22 µM Benzyl adenine, gelled with wheat starch as an agar alternative	Shoot Organogenesis	[21]
Leaf and Nodal segments	MS Medium with 0.3 mg/L 2,4-D and 0.1–0.3 mg/L BAP	Somatic Embryogenesis	[22]
Shoot tips	MS medium supplemented with 225.24 µM BAP	Shoot Organogenesis	[23]
Hypocotyl	MS Medium with 0.25 mg/L 2,4-D + 1 mg/L TDZ	Somatic Embryogenesis	[24]
Axillary bud	0.1 mg/L BAP + Liquid basal medium in temporary immersion system (TIS) using bioreactor Plantform^TM^	In vitro micropropagation (Large scale shoot production)	[25]

**Table 2 plants-13-00151-t002:** Somatic embryo induction in *L. barbarum* leaf explants’ response to picloram and 2,4-D.

Treatments	Picloram (µM)	2,4-D (µM)	Responsive Explant	Somatic Embryo Induction
1	Control		-	-
2	0.1		Leaf	Somatic embryos
3	1.0		Leaf	Somatic embryos
4	10.0		Leaf and Leaf with petiole	Callus
5		0.1	Leaf	Somatic embryos
6		1.0	Leaf and Leaf with petiole	Callus
7		10.0	leaf	Callus
8	0.05	0.05	Leaf	Somatic embryos
9	0.5	0.5	leaf	Callus
10	5.0	5.0	leaf	Callus

**Table 3 plants-13-00151-t003:** Effect of plant growth regulators on somatic embryo differentiation from leaf explants of *L. barbarum*.

Treatments	Picloram (µM)	2,4-D (µM)	BAP (µM)	Mean Number of Somatic Embryos per Explant
1 (Control)	0.00	0.00	0.00	0.00 ± 0.00 ^a^
2	0.1		1.0	1.35 ± 0.13 ^ef^
			5.0	5.45 ± 0.40 ^c^
			10.0	5.65 ± 0.63 ^c^
3	1.0		1.0	7.80 ± 0.50 ^b^
			5.0	4.00 ± 0.47 ^d^
			10.0	10.40 ± 0.47 ^a^
4		0.1	1.0	4.05 ± 0.51 ^d^
			5.0	1.65 ± 0.34 ^ef^
			10.0	2.55 ± 0.41 ^e^
5	0.05	0.05	1.0	1.15 ± 0.29 ^f^
			5.0	1.60 ± 0.31 ^ef^
			10.0	1.10 ± 0.29 ^f^

Each value represents mean ± SE. Values with the same superscript are not significantly different at 5% probability level according to DMRT. Best treatment response is in grey shade.

**Table 4 plants-13-00151-t004:** Schematic of optimized methodology detailing stages during somatic embryogenesis of *L. barbarum*.

No.	Stages	Procedure and Media Composition	Conditions and Durations	Results
1	Culture establishment	Leaves and leaves with petioles explants were immersed in 70% (*v*/*v*) ethanol for 1 min, followed by immersion in 15% sodium hypochlorite solution for 10 min and then rinsed three times.	All steps were performed under laminar airflow cabinet.	Plant materials were surface sterilized.
2	Callus induction	MS basal medium [12] Picloram, 2,4-D and Picloram + 2,4-D (0.1 µM, 1.0 µM and 10.0 µM).	25 ± 2 °C Darkness 5 weeks (1 subculture). Developing cultures were examined at weekly intervals using a microscope (Nikon SMZ 1270).	Nodular and friable embryogenic calluses were initiated.
3	Stabilization of embryogenic callus	MS basal medium BAP (1.0 µM, 5.0 µM and 10.0 µM).	25 ± 2 °C; 14 h light 60 μmol m^−2^ s ^−1^; 6 weeks (2 subcultures).	Embryogenic calluses were selected for further proliferation. Somatic embryos were differentiated.
4	Development of somatic embryos	MS basal medium, 30 g L^−1^ sucrose.	25 ± 2 °C; 14 h light 60 μmol m^−2^ s^−1^; 7–8 weeks (3 subcultures).	Calluses with plenty of embryos were obtained.
5	Germination of somatic embryos	MS basal medium, 30 g L^−1^ sucrose.	25 ± 2 °C; 14 h light at 60 μmol m^−2^ s ^−1^ until plantlet regeneration.	Further conversion of embryos and plantlet regeneration was obtained. Germinated bipolar embryos with well-formed shoots and roots were used.
6	Acclimatization of plantlets	Germinated plants were washed from agar and planted into humidity-controlled boxes. Promix BX, Premier Horticulture Inc., Pennsylvania was used.	25 ± 2 °C; 16 h light 60 μmol m^−2^ s^−1^; 21 days (about 3 weeks) until new growth was visible.	Plants were acclimatized into humidity-controlled boxes.

## Data Availability

All data are presented within the article.

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
