# Peer review of "Effect of Picloram and Desiccation on the Somatic Embryogenesis of Lycium barbarum L."

_plants, 2024, doi:10.3390/plants13020151_

Round 1

Reviewer 1 Report

Comments and Suggestions for Authors

Dear Authors,

The presented topic is interesting and useful for the fruit-growing sector.

There are a few points that should be addressed by the authors:

-        First of all, the authors should pay attention to the Latin names of different species of plants all over the paper, including in the title.

-        there are a lot of words split, especially at the beginning of the paper

-        It could indicate more precisely the source of plant material  Fort Valley State University... state, country...

-        The discussions highlight the results and demonstrate the possibility of reproducible, and rapid methods for in-vitro multiplication of Lycium barbarum L.

The paper reports for the first time the role of Picloram and desiccation on the induction of somatic embryos in Goji berry. Even if it needs further investigations to optimize the most suitable desiccation treatment for mass multiplication, this protocol could be a possible aid to the mass propagation of the Goji plants.

Thank you!

Author Response

Dear Reviewers,

Thank you for the feedback on our manuscript. We appreciate the time and effort you all have taken to review our work. We have made the changes you suggested and have added your suggestions to the manuscript. Author’s comments are in black. Please find below the responses to all the comments-

Reviewer 1:

  1. First of all, the authors should pay attention to the Latin names of different species of plants all over the paper, including in the title.

Authors Comments: We have made changes as per the reviewer’s suggestion. In fact, all the scientific names in the submitted word document were italicized. However, we noticed that there may be an issue with the formatting by Journal. We would like to bring this to the attention of the Journal editor.

  1. there are a lot of words split, especially at the beginning of the paper

Authors Comments: We have made changes as per the reviewer’s comments. The manuscript has undergone changes due to formatting issues, including word splits.

  1. It could indicate more precisely the source of plant material Fort Valley State University... state, ..

Authors Comments: We have added more information as per the reviewer’s suggestion. Kindly, refer to line 277.

Reviewer 2 Report

Comments and Suggestions for Authors

 In this study, the authors investigated the effect of different auxins and cytokinins on callus and somatic embryo formation of goji berry plants. For the first time, the effect of picloram and desiccation treatments on the regeneration of this plant was investigated. Although there are quite a lot of studies in the literature, this study is suitable for publication in the journal Plants because of its novelty. There are also minor points that need to be corrected.

1. All scientific species names should be italicized

2. Table 3 should be updated because it is not clear which concentration is effective.

Author Response

Dear Reviewers,

Thank you for the feedback on our manuscript. We appreciate the time and effort you all have taken to review our work. We have made the changes you suggested and have added your suggestions to the manuscript. Author’s comments are in black. Please find below the responses to all the comments-

Reviewer 2:

  1. All scientific species names should be italicized

Authors Comments: Kindly refer to reviewer 1 (Comment 1).

  1. Table 3 should be updated because it is not clear which concentration is effective.

Authors Comments: Best treatment has been highlighted for more clarity in Table 3. Also, a Note has been added in the line 148.

Reviewer 3 Report

Comments and Suggestions for Authors

The manuscript titled “Effect of Picloram and desiccation on the somatic embryogenesis of Lycium barbarum L.” by Khatari and Joshee demonstrates an effective in vitro propagation protocol of Goji berries, one of the plants producing valuable fruits (nutritional, antioxidant, supporting the treatment of many diseases). The authors found a positive effect of Picloram at a concentration of 1.0 µM and drying at room temperature for 9 and 12 hours on the induction of somatic embryos from leaf explants. They showed that 100% of the plants acclimatized to greenhouse conditions.

This is one of the few protocols enabling somatic embryogenesis in an extremely valuable plant. The results obtained by the authors show that this species can be effectively propagated using this method. In the future, after automating the process of somatic embryogenesis, this would enable intensive production of somatic L. fruits plants. Therefore, I believe that the manuscript I am reviewing deserves to be published in Plants, taking into account the comments and suggestions presented below.

Abstract

I suggest adding information in the summary that well-developed plants have been obtained and are capable of acclimatizing in ex vitro conditions.

Page 1, line 10

Correct 1 µM into 1.0 µM

Results and Discussion

The discussion should explain the differences between the auxins used in terms of their use in in vitro cultures. Did the type of auxin used (Picloram, 2,4-D) significantly affect the number of abnormal embryos obtained? Another issue is to provide a potential reason for the possibility of inducing somatic embryogenesis from leaves, but the lack of a positive reaction from the second type of explant. In my opinion, the discussion regarding plant acclimatization could also be expanded a bit. What results did other authors obtain at this stage? Is the result obtained by the authors the best compared to other studies?

Page 7, line 187

Write plant names in italics, applies to the entire text.

Page 7, line 193-196

It is worth mentioning how many plants were obtained and acclimatized in climate-controlled boxes. And some information, how did they grow under greenhouse conditions (their high, and general condition)

Page 9, line 229

An additional table showing these results should be included.

Materials and methods

Page 10, line 279

Correct the temperature record.

Page 12, line 306

What was the light intensity at this stage?

Tables descriptions

Table 1

In the table description add that the species regeneration concerns in vitro culture.

Table 2

In the table description add that results for both leaves and leaves with petiole explants are presented here.

Table 3

I suggest more precisely descript this Table, for example:

Table 3. Effect of plant growth regulator on somatic embryos differentiation from explants of L. barbarum.

Table 4

I suggest to change the description of the table in this manner:

Table 4. Methodology used and results obtained at individual stages of somatic embryogenesis of L. barbarum.

Figures

Figure 1.

Legends

Leaf with petiole instead Petiole

It would be advisable to add whiskers to the pots.

Figure 3.

The scale should be placed on photos A and C

Final comment

The text contains a few editing errors that need to be corrected (see Page 2, line 52 bev-erages etc. on next pages).

Author Response

Dear Reviewers,

Thank you for the feedback on our manuscript. We appreciate the time and effort you all have taken to review our work. We have made the changes you suggested and have added your suggestions to the manuscript. Author’s comments are in black. Please find below the responses to all the comments-

Reviewer 3:

  1. Abstract: I suggest adding information in the summary that well-developed plants have been obtained and are capable of acclimatizing in ex vitro conditions.

Authors Comments: The abstract has been updated with additional information as per the reviewer’s recommendations. Please refer to line 19 for more information.

Page 1, line 10

Correct 1 µM into 1.0 µM

Authors Comments: We have made changes as per the reviewer’s comments. Please refer to lines 124- 127. We have also corrected it for other concentrations as well.

  1. Results and Discussion: The discussion should explain the differences between the auxins used in terms of their use in in vitro cultures.

Authors Comments: More information has been added as per reviewer’s suggestion. Kindly refer to the lines 111-114.

Did the type of auxin used (Picloram, 2,4-D) significantly affect the number of abnormal embryos obtained?

Authors Comments: No abnormal embryos were observed when auxins (Picloram and 2,4-D) used at a low concentration and incubated for no more than 5 weeks.

Another issue is to provide a potential reason for the possibility of inducing somatic embryogenesis from leaves, but the lack of a positive reaction from the second type of explant.

Authors Comments: Petiole alone has demonstrated both positive and negative responses to somatic embryogenesis. The addition of petiole to the leaf may impact the induction of somatic embryos. However, we can not include this information in the text as it requires separate experimentation. Nonetheless, we have included a few reasons that other researchers have reported. Kindly refer to line 106-110.

In my opinion, the discussion regarding plant acclimatization could also be expanded a bit. What results did other authors obtain at this stage? Is the result obtained by the authors the best compared to other studies?

Authors Comments: Other authors have reported variable response ranging from 80 to 95%. More information has been added to the acclimatization section as per reviewer suggestions. Please refer line 203– 208.

             Page 7, line 187

Write plant names in italics, applies to the entire text.

Authors Comments: We have made Changes as per reviewers comments. Please refer reviewer 1 (comment 1). 

  1. Page 7, line 193-196

It is worth mentioning how many plants were obtained and acclimatized in climate-controlled boxes. And some information, how did they grow under greenhouse conditions (their high, and general condition)

Authors Comments: We have added more information as per reviewers’ comments. Kindly refer to the line 196- 203.

  1. Page 9, line 229

An additional table showing these results should be included.

Authors Comments: We have added the graphical representation of the results as per reviewer’s suggestion. Kindly refer to figure 6.

  1. Materials and methods

Page 10, line 279

Correct the temperature record.

Authors Comments: We have corrected the temperature record. The manuscript has undergone changes due to formatting issues, including word splits, scientific names non-Italics and incorrect temperature record. We have addressed this issue to the Editor as well.

  1. Page 12, line 306

What was the light intensity at this stage?

Authors Comments: Light Intensity has been added. Kindly refer to line 331.

  1. Tables descriptions  Table 1

In the table description add that the species regeneration concerns in vitro culture.

Authors Comments: We have reframed the sentence as per reviewer’s suggestion. Table 1 provides a review of the protocols used by other researchers and does not address regeneration concerns. However, if you feel that this description should be included, we can accommodate your suggestions. Kindly refer to line 77.

  1. Table 2

In the table description add that results for both leaves and leaves with petiole explants are presented here.

  Authors Comments: We have modified the caption per reviewer’s suggestion. Kindly refer to the line 88.

  1. Table 3

I suggest more precisely descript this Table, for example:

Table 3. Effect of plant growth regulator on somatic embryos differentiation from explants of L. barbarum.

Authors Comments: We have modified the caption as per reviewer’s suggestion. Kindly refer to line 145-141.

  1. Table 4

I suggest to change the description of the table in this manner:

Table 4. Methodology used and results obtained at individual stages of somatic embryogenesis of L. barbarum.

Authors Comments: We have modified the caption as per reviewer’s suggestion. Kindly refer to line 1ine 309.

  1. Figures

 Figure 1.

Legends

Leaf with petiole instead Petiole

It would be advisable to add whiskers to the pots.

Authors Comments: Running the software again for all the treatments is necessary to add the whiskers. However, this process is time consuming and will cause delays. Details have been provided in the note. Kindly refer to line 101-103.

  1. Figure 3.

The scale should be placed on photos A and C

Authors Comments: For Photo A- we have added scale as per reviewer’s suggestion. Kindly refer to figure 3.

For Photo C- we have added the note. Kindly refer to line 144.

  1. Final comment

The text contains a few editing errors that need to be corrected (see Page 2, line 52 bev-erages etc. on next pages).

Authors Comments: Kindly refer to the comment1 in reviewer1 section.

Round 2

Reviewer 3 Report

Comments and Suggestions for Authors

I would like to thank the authors for including most of the reviewer's comments to the text. I have no objection to the current content and form of the manuscript.